# WHAT MATTERS TO *You*? TOWARDS VISUAL REPRESENTATION ALIGNMENT FOR ROBOT LEARNING

**Ran Tian[1], Chenfeng Xu[1], Masayoshi Tomizuka[1], Jitendra Malik[1], Andrea Bajcsy[2]**
[1]UC Berkeley    [2]Carnegie Mellon University

## ABSTRACT

When operating in service of people, robots need to optimize rewards aligned with end-user preferences. Since robots will rely on raw perceptual inputs, their rewards will inevitably use *visual* representations. Recently there has been excitement in using representations from pre-trained visual models, but key to making these work in robotics is fine-tuning, which is typically done via proxy tasks like dynamics prediction or enforcing temporal cycle-consistency. However, all these proxy tasks bypass the human's input on what matters to *them*, exacerbating spurious correlations and ultimately leading to behaviors that are misaligned with user preferences. In this work, we propose that robots should leverage human feedback to *align* their visual representations with the end-user and disentangle what matters for the task. We propose **R**epresentation-**A**ligned **P**reference-based **L**earning (**RAPL**), a method for solving the visual representation alignment problem and visual reward learning problem through the lens of preference-based learning and optimal transport. Across experiments in X-MAGICAL and in robotic manipulation, we find that RAPL's reward consistently generates preferred robot behaviors with high sample efficiency, and shows strong zero-shot generalization when the visual representation is learned from a different embodiment than the robot's.

## 1 INTRODUCTION

Imagine that a robot manipulator is tasked with cleaning your kitchen countertop. To be successful, its reward function should model your preferences: what you think an orderly countertop looks like (e.g., plates should be stacked by color), how objects should be handled during cleaning (e.g., cups should be moved one-at-a-time but food scraps should be pushed in large groups), and what parts of the countertop should be avoided (e.g., always stay away from the expensive espresso machine).

A long-standing approach to this problem has been through inverse reinforcement learning (IRL), where the robot infers a reward from demonstrations. Fundamental work in IRL (Abbeel & Ng, 2004; Ziebart et al., 2008) argues for matching features between the expert and the learner during reward inference, more formally known as optimizing Integral Probability Metrics (IPMs) (Sun et al., 2019; Swamy et al., 2021). While there are many ways to do this matching, optimal transport has recently been used to optimize IPMs defined on high-dimensional feature spaces in a principled manner (Xiao et al., 2019; Dadashi et al., 2021; Papagiannis & Li, 2022; Luo et al., 2023).

The question remains, what are good features for the learner to match? While traditional work used hand-engineered features defined on low-dimensional states of the world that the robot could do state estimation on (e.g. object pose) (Levine et al., 2011; Finn et al., 2016), large vision models trained via self-supervised/unsupervised learning promise representations for robotics that directly operate on raw image inputs and capture more nuanced features automatically (e.g., object color and shape) (Xiao et al., 2022; Ma et al., 2023; Karamcheti et al., 2023). However, automatically extracting relevant visual features that incentivize preferred robot behavior and minimize spurious correlations remains an open challenge (Zhang et al., 2020). Previous works use proxy tasks during representation learning to imbue some prior human knowledge about attributes that are relevant (Brown et al., 2020), such as assuming access to action labels and doing behavior cloning (Haldar et al., 2023a;b) or performing temporal cycle-consistency learning (Dadashi et al., 2021). The former requires additional signals that may be hard to obtain (i.e., actions), while the latter assumes that

Figure 1: **Representation-Aligned Preference-based Learning** (**RAPL**), is an action-free visual representation learning method using easy-to-provide human preference feedback on video demos. Using the human preference triplets, the robot goes from paying attention to the end-effector ($\tilde{\phi}_H^0$) to paying attention to the objects and the goal region ($\tilde{\phi}_H^*$) at the end of alignment. The aligned representation is used to construct an optimal transport-based visual reward for robot behavior learning.

temporal consistency signal is sufficient to extract what matters to the task, potentially ignoring other features that matter to the demonstrator (e.g., avoiding undesired regions).

Since robots ultimately operate in service of people, recent works advocate that the robot should explicitly engage in a *representation alignment process* with the end-user. These works leverage user feedback such as human-driven feature selection (Bullard et al., 2018; Luu-Duc & Miura, 2019), interactive feature construction (Bobu et al., 2021; Katz et al., 2021) or similarity-queries (Bobu et al., 2023a) to learn human-centric representations. However, these approaches either operate on pre-defined feature sets or in low-dimensional state spaces (e.g., positions). In the visual domain, (Zhang et al., 2020) uses manually-defined reward signals to learn representations which disentangle high versus low reward images; unfortunately, when the main objective *is* reward learning in the first place, this approach is not feasible.

*Instead of hoping to extract human-centric visual representations via proxy tasks that bypass human input or require action labels per video frame, we propose that robots use easy-to-provide human preference feedback—where the human is asked to compare two or more video demonstrations—to align their visual representations with what matters to the end-user (Figure 1).*

We first formalize the visual representation alignment problem for robotics as a metric learning problem in the human's representation space. We then propose **R**epresentation-**A**ligned **P**reference-based **L**earning (**RAPL**), a tractable video-only method for solving the alignment problem and learning visual robot rewards via optimal transport. Across experiments in X-Magical (Zakka et al., 2022) and in robotic manipulation, we find that RAPL's reward consistently generates preferred robot behaviors with high sample efficiency, and shows strong zero-shot generalization when the visual representation is learned on a different embodiment than the robot's.

## 2 PROBLEM SETUP

**Human Policy.** We consider scenarios where the robot R wants to learn how to perform a task for human H. The human knows the desired reward $r^*$ which encodes their preferences for the task. The human acts via an approximately optimal policy $\pi_H^* : \mathcal{O} \to \mathcal{U}_H$ based on their underlying reward function. Instead of directly consuming the raw perceptual input, research suggests that humans naturally build visual representations of the world (Bonnen et al., 2021) that focus on task-relevant attributes (Callaway et al., 2021). We model the human's representation model as $\phi_H : \mathcal{O} \to \Phi_H$, mapping from the perceptual input $o \in \mathcal{O}$ to the human's latent space $\Phi_H$ which captures their task and preference-relevant features.

**Robot Policy.** We seek to learn a robot policy $\pi_R^* : \mathcal{O} \to \mathcal{U}_R$ which maps from image observations to actions that maximizes a reward:

$$\pi_R^* = \arg\max_{\pi_R} \mathbb{E}_{\mathbf{o} \sim p(\mathbf{o}|\pi_R)} \Big[ \sum_{t=0}^{\infty} \gamma^t \cdot r(\phi_R(o^t)) \Big], \tag{1}$$

where $\gamma \in [0, 1)$ is the discount factor and $\mathbf{o} = \{o^0, o^1, \ldots\}$ is the image observation trajectory induced by the robot's policy. The robot's reward $r$ relies on a visual representation, $\phi_R : \mathcal{O} \to \Phi_R$, which maps from the image observation $o \in \mathcal{O}$ to a lower-dimensional latent space, $\Phi_R$. In general, this could be hand-crafted, such as distances to objects from the agent's end-effector (Ziebart et al., 2008; Levine et al., 2011; Finn et al., 2016), or the output of an encoder pre-trained on large-scale

datasets (Chen et al., 2021; Ma et al., 2023). Before the robot can optimize for $\pi_R$, it is faced with two questions: what visual representation $\phi_R$ should it use to encode observations, and which reward $r$ should it optimize to align its behavior with $\pi_H^*$?

# 3 RAPL: REPRESENTATION-ALIGNED PREFERENCE-BASED LEARNING

In this work, we leverage pre-trained visual encoders but advocate that robots fine-tune them with *preference-based human feedback* to extract features that are relevant for how the end-user likes the task to be performed. Preference-based feedback, where a user is asked to compare two or more trajectories, has been shown to be easier for end-users to provide compared to direct labelling or giving near-optimal demonstrations (Wirth et al., 2017). However, we are the first to use preference-based human feedback to align pre-trained visual models with user preference for robot learning.

Once the robot has an aligned representation, what reward should it optimize to generate preferred behaviors? To reduce the sample complexity, we use optimal transport methods to design a visual reward (Villani et al., 2009), and focus the preference feedback exclusively on fine-tuning the representation. Since optimal transport methods are a way to optimize Integral Probability Metrics in a principled manner, the transport plan exactly yields a reward which maximizes feature matching between the learner and the expert *in our aligned representation space*.

In this section, we first formally state the visual representation alignment problem by drawing upon recent work in cognitive science (Sucholutsky & Griffiths, 2023) and then detail our approximate solution, **R**epresentation-**A**ligned **P**reference-based **L**earning (**RAPL**), through the lens of preference-based learning and optimal transport. In addition to this section's content, we also provide more details about comparisons between our work and previous work in Appendix A.2 and A.1.

## 3.1 THE VISUAL REPRESENTATION ALIGNMENT PROBLEM FOR ROBOTICS

We follow the formulation in Sucholutsky & Griffiths (2023) and bring this to the robot learning domain. Intuitively, visual representation alignment is defined as the degree to which the output of the robot's encoder, $\phi_R$, matches the human's internal representation, $\phi_H$, for the same image observation, $o \in \mathcal{O}$, during task execution. We utilize a triplet-based definition of representation alignment as in (Jamieson & Nowak, 2011) and (Sucholutsky & Griffiths, 2023).

**Definition 1 (Triplet-based Representation Space)** *Let $\mathbf{o} = \{o^t\}_{t=0}^T$ be a sequence of image observations over $T$ timesteps, $\phi : \mathcal{O} \to \Phi$ be a given representation model, and $\phi(\mathbf{o}) := \{\phi(o^0), \ldots, \phi(o^T)\}$ be the corresponding embedding trajectory. For some distance metric $d(\cdot, \cdot)$ and two observation trajectories $\mathbf{o}^i$ and $\mathbf{o}^j$, let $d(\phi(\mathbf{o}^i), \phi(\mathbf{o}^j))$ be the distance between their embedding trajectories. The triplet-based representation space of $\phi$ is:*

$$S_\phi = \left\{ (\mathbf{o}^i, \mathbf{o}^j, \mathbf{o}^k) : d(\phi(\mathbf{o}^i), \phi(\mathbf{o}^j)) < d(\phi(\mathbf{o}^i), \phi(\mathbf{o}^k)), \mathbf{o}^{i,j,k} \in \Xi \right\}, \quad (2)$$

*where $\Xi$ is the set of all possible image trajectories for the task of interest.*

Intuitively, this states that the visual representation $\phi$ helps the agent determine how similar two videos are in a lower-dimensional space. For all possible triplets of videos that the agent could see, it can determine which videos are more similar and which videos are less similar using its embedding space. The set $S_\phi$ contains all such similarity triplets.

**Definition 2 (Visual Representation Alignment Problem)** *Recall that $\phi_H$ and $\phi_R$ are the human and robot's visual representations respectively. The representation alignment problem is defined as learning a $\phi_R$ which minimizes the difference between the two agents' representation spaces, as measured by a function $\ell$ which penalizes divergence between the two representation spaces:*

$$\min_{\phi_R} \ell(S_{\phi_R}, S_{\phi_H}). \quad (3)$$

## 3.2 REPRESENTATION INFERENCE VIA PREFERENCE-BASED LEARNING

Although this formulation sheds light on the underlying problem, solving Equation 3 exactly is impossible since the functional form of the human's representation $\phi_H$ is unavailable and the set

$S_{\phi_H}$ is infinite. Thus, we approximate the problem by constructing a subset $\tilde{S}_{\phi_H} \subset S_{\phi_H}$ of triplet queries. Since we seek a representation that is relevant to the human's preferences, we ask the human to rank these triplets based on their *preference-based* notion of similarity (e.g., $r^*(\phi_H(\mathbf{o}^i)) > r^*(\phi_H(\mathbf{o}^j)) > r^*(\phi_H(\mathbf{o}^k)) \implies \mathbf{o}^i \succ \mathbf{o}^j \succ \mathbf{o}^k$). With these rankings, we implicitly learn $\phi_H$ via a neural network trained on these triplets.

We interpret a human's preference over the triplet $(\mathbf{o}^i, \mathbf{o}^j, \mathbf{o}^k) \in \tilde{S}_{\phi_H}$ via the Bradley-Terry model (Bradley & Terry, 1952), where $\mathbf{o}^i$ is treated as an anchor and $\mathbf{o}^j, \mathbf{o}^k$ are compared to the anchor in terms of similarity as in Equation 2:

$$\mathbb{P}(\mathbf{o}^i \succ \mathbf{o}^j \succ \mathbf{o}^k \mid \phi_H) \approx \frac{e^{-d(\phi_H(\mathbf{o}^i),\, \phi_H(\mathbf{o}^j))}}{e^{-d(\phi_H(\mathbf{o}^i),\, \phi_H(\mathbf{o}^j))} + e^{-d(\phi_H(\mathbf{o}^i),\, \phi_H(\mathbf{o}^k))}}. \tag{4}$$

A natural idea would be to leverage the human's preference feedback to do direct reward prediction (i.e., model both $d$ and $\phi_H$ via a single neural network to approximate $r^*$), as is done in traditional preference-based reward learning (Christiano et al., 2017). However, we find empirically (Section 5.2) that directly learning a high-quality visual reward from preference queries requires a prohibitive amount of human feedback that is unrealistic to expect from end-users (Brown et al., 2019; Bobu et al., 2023b).

Instead, we focus all the preference feedback on just representation alignment. But, this raises the question what distance measure $d$ should we use? In this work, we use optimal transport as a principled way to measure the feature matching between any two videos. For any video $\mathbf{o}$ and for a given representation $\phi$, let the induced empirical embedding distribution be $\rho = \frac{1}{T} \sum_{t=0}^{T} \delta_{\phi(o^t)}$, where $\delta_{\phi(o^t)}$ is a Dirac distribution centered on $\phi(o^t)$. Optimal transport finds the optimal coupling $\mu^* \in \mathbb{R}^{T \times T}$ that transports one embedding distribution, $\rho_i$, to another video embedding distribution, $\rho_j$, with minimal cost. This approach has a well-developed suite of numerical solution techniques (Peyré et al., 2019) which we leverage in practice (for details on this see App. A.3).

Our final optimization is a maximum likelihood estimation problem:

$$\tilde{\phi}_H := \max_{\phi_H} \sum_{(\mathbf{o}^i, \mathbf{o}^j, \mathbf{o}^k) \in \tilde{S}_{\phi_H}} \mathbb{P}(\mathbf{o}^i \succ \mathbf{o}^j \succ \mathbf{o}^k \mid \phi_H). \tag{5}$$

Since the robot seeks a visual representation that is aligned with the human's, we set: $\phi_R := \tilde{\phi}_H$.

### 3.3 Preference-Aligned Robot Behavior via Optimal Transport

Given our aligned visual representation, we seek a robot policy $\pi_R$ whose behavior respects the end-user's preferences. Traditional IRL methods (Abbeel & Ng, 2004; Ziebart et al., 2008) are built upon matching features between the expert and the learner. Leveraging this insight, we use optimal transport methods since the optimal transport plan is equivalent to defining a reward function that encourages this matching (Kantorovich & Rubinshtein, 1958).

Specifically, we seek to match the *embedded* observation occupancy measure induced by the robot's policy $\pi_R$, and the embedded observation occupancy measure of a human's preferred video demonstration, $\mathbf{o}_+$. Thus, the optimal transport plan yields the reward which is optimized in Equation 1:

$$r(o_R^t; \phi_R) = -\sum_{t'=1}^{T} c\big(\phi_R(o_R^t), \phi_R(o_+^{t'})\big) \mu_{t,t'}^*, \tag{6}$$

where $\mu^*$ is the optimal coupling between the empirical embedding distributions induced by the robot and the expert. This reward has been successful in prior vision-based robot learning (Haldar et al., 2023b;a; Guzey et al., 2023) with the key difference in our setting being that we use RAPL's aligned visual representation $\phi_R$ for feature matching (for details on the difference between our approach and prior OT based visual reward see Appendix A.1).

## 4 Experimental Design

We design a series of experiments to investigate RAPL's ability to learn visual rewards and generate preferred robot behaviors.

**Preference Dataset:** $\tilde{S}_{\phi_{\mathrm{H}}}$**.** While the ultimate test is learning from real end-user feedback, in this work we use a simulated human model as a first step. This allows us to easily ablate the size of the preference dataset, and gives us privileged access to $r^*$ for direct comparison. In all environments, the simulated human constructs the preference dataset $\tilde{S}_{\phi_{\mathrm{H}}}$ by sampling triplets of videos uniformly at random from the set[1] of video observations $\tilde{\Xi} \subset \Xi$, and then ranking them with their reward $r^*$ as in Equation 4.

**Independent & Dependent Measures.** Throughout our experiments we vary the *visual reward signal* used for robot policy optimization and the *preference dataset size* used for representation learning. We measure robot task success as a binary indicator of if the robot completed the task with high reward $r^*$.

**Controlling for Confounds.** Our ultimate goal is to have a visual robot policy, $\pi_{\mathrm{R}}$, that takes as input observations and outputs actions. However, to rigorously compare policies obtained from different visual rewards, we need to disentangle the effect of the reward signal from any other policy design choices, such as the input encoders and architecture. To have a fair comparison, we follow the approach from (Zakka et al., 2022; Kumar et al., 2023) and input the privileged ground-truth state into all policy networks, but vary the visual reward signal used during policy optimization. Across all methods, we use an identical reinforcement learning setup and Soft-Actor Critic for training (Haarnoja et al., 2018) with code base from (Zakka et al., 2022).

In Section 5, we first control the agent's embodiment to be consistent between both representation learning and robot optimization (e.g., assume that the robot shows video triplets of itself to the human and the human ranks them). In Section 6, we relax this assumption and consider the more realistic cross-embodiment scenario where the representation learning is performed on videos of a different embodiment than the robot's. For all policy learning experiments, we use 10 expert demonstrations as the demonstration set $\mathcal{D}_+$ for generating the reward (more details in Appendix A.3).

## 5 RESULTS: FROM REPRESENTATION TO BEHAVIOR ALIGNMENT

We first experiment in the toy X-Magical environment (Zakka et al., 2022), and then move to the realistic IsaacGym simulator.

### 5.1 X-MAGICAL

**Tasks.** We design two tasks inspired by kitchen countertop cleaning. The robot always has to push objects to a goal region (e.g., trash can), shown in pink at the top of the scene in Figure 2. In the **avoiding** task, the end-user prefers that the robot and objects never enter an *off-limits zone* during pushing (blue box in left Figure 2). In the **grouping** task, the end-user prefers that objects are always pushed *efficiently together* (instead of one-at-a-time) towards the goal region (center, Figure 2).

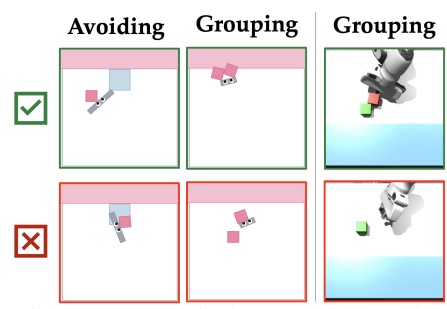

Figure 2: X-Magical & IsaacGym tasks.

**Privileged State & Reward.** For **avoiding**, the true state $s$ is 7D: planar robot position ($p_{\mathrm{R}} \in \mathbb{R}^2$) and orientation ($\theta_{\mathrm{R}}$), planar position of the object ($p_{\mathrm{obj}} \in \mathbb{R}^2$), distance between goal region and object, ($d_{\mathrm{obj2goal}}$), and distance between the off-limits zone and the object ($d_{\mathrm{obs2obj}}$). The human's reward is: $r^*_{\mathrm{avoid}}(s) = -d_{\mathrm{goal2obj}} - 2 \cdot \mathbb{I}(d_{\mathrm{obs2obj}} < d_{\mathrm{safety}})$, where $d_{\mathrm{safety}}$ is a safety distance and $\mathbb{I}$ is an indicator function giving 1 when the condition is true. For **grouping**, the state is 9D: $s := (p_{\mathrm{R}}, \theta_{\mathrm{R}}, p_{\mathrm{obj^1}}, p_{\mathrm{obj^2}}, d_{\mathrm{goal2obj^1}}, d_{\mathrm{goal2obj^2}})$. The human's reward is: $r^*_{\mathrm{group}}(s) = -\max(d_{\mathrm{goal2obj^1}}, d_{\mathrm{goal2obj^2}}) - ||p_{\mathrm{obj^1}} - p_{\mathrm{obj^2}}||_2$.

**Baselines.** We compare our visual reward, **RAPL**, against (1) **GT**, an oracle policy obtained under $r^*$, (2) **RLHF**, which is vanilla preference-based reward learning (Christiano et al., 2017; Brown

---

[1]To minimize the bias of this set on representation learning, we construct $\tilde{\Xi}$ such that the reward distribution of this set under $r^*$, is approximately uniform. Future work should investigate the impact of this set further, e.g., (Sadigh et al., 2017).

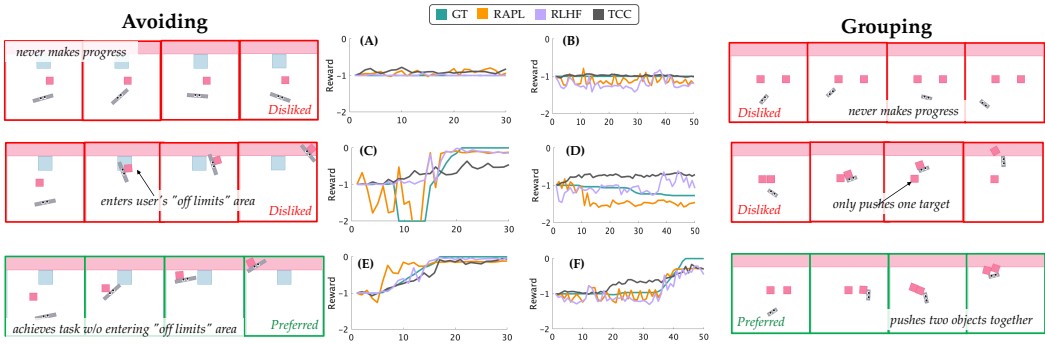

Figure 3: **X-Magical.** (left & right) examples of preferred and disliked videos for each task. (center) reward associated with each video under each method. RAPL's predicted reward follows the GT pattern: low reward when the behavior are disliked and high reward when the behavior are preferred. RLHF and TCC assign high reward to disliked behavior (e.g., (D)).

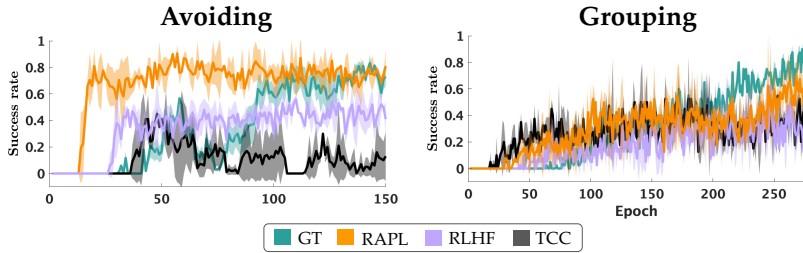

Figure 4: **X-Magical.** Policy evaluation success rate during policy learning. Colored lines are the mean and variance of the evaluation success rate. RAPL can match GT in the **avoiding** task and outperforms baseline visual rewards in **grouping** task.

et al., 2019) that directly maps an image observation to a scalar reward, and (3) TCC (Zakka et al., 2022; Kumar et al., 2023) which finetunes a pre-trained encoder via temporal cycle consistency constraints using 500 task demonstrations and then uses L2 distance between the current image embedding and the goal image embedding as reward. We use the same preference dataset with 150 triplets for training RLHF and RAPL.

**Visual backbone.** We use the same setup as in (Zakka et al., 2022) with the ResNet-18 backbone pre-trained on ImageNet. The original classification head is replaced with a linear layer that outputs a 32-dimensional vector as our embedding space, $\Phi_R := \mathbb{R}^{32}$. The TCC representation model is trained with 500 demonstrations using the code from (Zakka et al., 2022). Both RAPL and RLHF only fine-tune the last linear layer. All representation models are frozen during policy learning.

**Hypothesis**. *RAPL is better at capturing preferences beyond task progress compared to direct reward prediction RLHF or TCC visual reward, yielding higher success rate.*

**Results.** Figure 3 shows the rewards over time for three example video observations in the **avoid** (left) and **group** task (right). Each video is marked as preferred by the end-user's ground-truth reward or disliked. Across all examples, RAPL's rewards are highly correlated with the GT rewards: when the behavior in the video is disliked, then reward is low; when the behavior is preferred, then the reward is increasing. TCC's reward is correlated with the robot making spatial progress (i.e., plot (E) and (F) where observations get closer to looking like the goal image), but it incorrectly predicts high reward when the robot makes spatial progress but violates the human's preference ((C) and (D) in Figure 3). RLHF performs comparably to RAPL, with slight suboptimality in scenarios (C) and (D). Figure 4 shows the policy evaluation success rate during RL training with each reward function (solid line is the mean, shaded area is the standard deviation, over 5 trials with different random seeds.). Across all environments, RAPL performs comparably to GT (**avoid** success: $\approx 80\%$, **group** success: $\approx 60\%$) and significantly outperforms all baselines with better sample efficiency, RAPL takes 10 epochs to reach 70% success rate in the **avoid** task (GT requires 100) and takes 100 epochs to reach 40% success rate in the **avoid** task (GT requires 150), supporting our hypothesis.

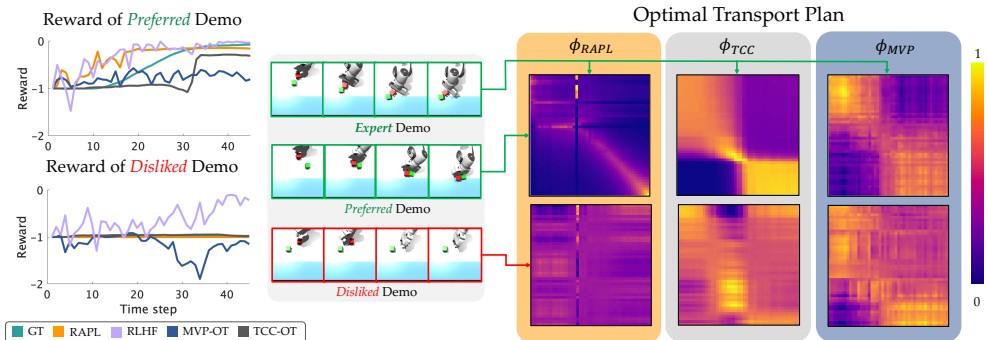

Figure 5: **Manipulation.** (center) Expert, preferred, and disliked video demo. (left) reward associated with each video under each method. RAPL's predicted reward follows the GT pattern. RLHF assigns high reward to disliked behavior. (right) OT coupling for each representation. Columns are embedded frames of expert demo. Rows of top matrices are embedded frames of preferred demo; rows of bottom matrices are embedded frames of disliked demo. Peaks exactly along the diagonal indicate that the frames of the two videos are aligned in the latent space; uniform values in the matrix indicate that the two videos cannot be aligned (i.e., all frames are equally "similar" to the next). RAPL matches this structure: diagonal peaks for expert-and-preferred and uniform for expert-and-disliked, while baselines show diffused values no matter the videos being compared.

## 5.2 ROBOT MANIPULATION

In the X-Magical toy environment, RAPL outperformed progress-based visual rewards, but direct preference-based reward prediction was a competitive baseline. Moving to the more realistic robot manipulation environment, we want to 1) disentangle the benefit of our fine-tuned representation from the optimal transport reward structure, and 2) understand if our method still outperforms direct reward prediction in a more complex environment?

**Task, Privileged State & Reward.** We design a robot manipulation task in the IsaacGym physics simulator (Makoviychuk et al., 2021). We replicate the tabletop **grouping** scenario, where a Franka robot arm needs to learn that the end-user prefers objects be pushed *efficiently together* (instead of one-at-a-time) to the goal region (light blue region in right of Figure 2). The state $s$ is 18D: robot proprioception ($\theta_{\text{joints}} \in \mathbb{R}^{10}$), 3D object positions ($p_{\text{obj}^{1,2}}$), and object distances to goal ($d_{\text{goal2obj}^{1,2}}$). The **grouping** reward is identical as in Section 5.1.

**Baselines.** In addition to comparing **RAPL** against (1) **GT** and (2) **RLHF**, we ablate the *representation model* but control the *visual reward structure*. We consider five additional baselines that all use optimal transport-based reward but operate on different representations: (3) **MVP-OT** which learns image representation via masked visual pre-training; (4) **Fine-Tuned-MVP-OT**, which fine-tunes MVP representation using images from the task environment; (5) **R3M-OT**, which is a ResNet-18 encoder (Nair et al., 2022) pre-trained on the Ego4D data set (Grauman et al., 2022) via a learning objective that combines time contrastive learning, video-language alignment, and a sparsity penalty; (6): **ImageNet-OT**, which is a ResNet-18 encoder pre-trained on ImageNet; (7) **TCC-OT** (Dadashi et al., 2021) which embeds images via the TCC representation trained with 500 task demonstrations. We use the same preference dataset with 150 triplets for training RLHF and RAPL.

**Visual model backbone.** All methods except MVP-OT and Fine-Tuned-MVP-OT share the same ResNet-18 visual backbone and have the same training setting as the one in the X-Magical experiment. MVP-OT and Fine-Tuned-MVP-OT use a off-the-shelf visual transformer (Xiao et al., 2022) pre-trained on the Ego4D data set (Grauman et al., 2022). All representation models are frozen during robot policy learning.

**Hypotheses**. **H1:** *RAPL's higher success rate is driven by its aligned visual representation.* **H2:** *RAPL outperforms RLHF with lower amounts of human preference queries.*

**Results: Reward prediction & policy learning.** In the center of Figure 5 we show three video demos: an expert video demonstration, a preferred video, and a disliked video. On the right of Figure 5, we visualize the optimal transport plan comparing the expert video to the disliked and preferred videos under visual representations $\phi_{\text{RAPL}}$, $\phi_{\text{TCC-OT}}$, $\phi_{\text{MVP-OT}}$. Intuitively, peaks exactly

along the diagonal indicate that the frames of the two videos are aligned in the latent space; uniform values in the matrix indicate that the two videos cannot be aligned (i.e., all frames are equally "similar" to the next). **RAPL**'s representation induces precisely this structure: diagonal peaks when comparing two preferred videos and uniform when comparing a preferred and disliked video. Interestingly, we see diffused peak regions in all transport plans under both **TCC-OT** and **MVP-OT** representations, indicating their representations struggle to align preferred behaviors and disentangle disliked behaviors. This is substantiated by the left of Figure 5, which shows the learned reward over time of preferred video and disliked video. Across all examples, **RAPL** rewards are highly correlated to **GT** rewards while baselines struggle to disambiguate, supporting **H1**. A more detailed quantitative analysis on the relationship between the learned visual reward and the end-user's ground-truth reward is shown in Appendix A.8.

Figure 6 shows the policy evaluation history during RL training with each reward function. We see **RAPL** performs comparably to **GT** (succ. rate: $\approx 70\%$) while all baselines struggle to achieve a success rate of more than $10\%$. It's surprising that **RLHF** fails in a more realistic environment since its objective is similar to ours, but without explicitly considering representation alignment. To further investigate this, we apply a linear probe on the final embedding and visualize the image heatmap of what each method's final embedding pays attention to in Figure 11 in the Appendix. **RLHF** is biased towards paying attention to irrelevant areas that can induce spurious correlations; in contrast **RAPL** focuses on the task-relevant objects and the goal region.

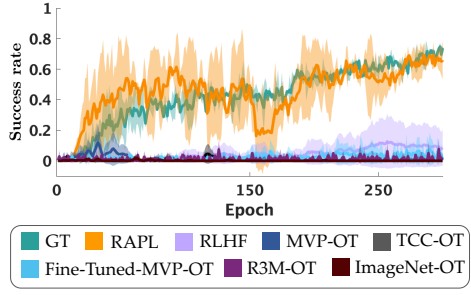

Figure 6: **Manipulation.** Success during robot policy learning for each visual reward.

**Results: Sample complexity.** We further study the sensitivity of RAPL and RLHF to the preference query dataset size. For RLHF, we double the preference query dataset during reward model training to 300 queries (detailed in App. A.5). Policy performance is improved, indicating that with more feedback data, preference-based reward prediction could yield an aligned policy. Nevertheless, **RAPL** outperforms **RLHF** by $75\%$ with $50\%$ less training data, supporting **H2**. While all the RAPL results above used 150 preference queries to train the representation, we also train a visual representation with 100, 50, and 25 preference queries. Figure 7 shows the success rate of the robot manipulation policy for each version of RAPL. We note that **RAPL** achieves a $45\%$ success rate even when trained on only 25 preference queries.

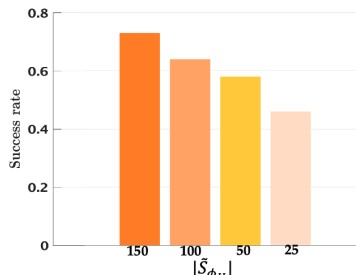

Figure 7: **Manipulation.** **RAPL** sample efficiency analysis. RAPL can achieve $\approx 45\%$ success rate with only 25 preference queries.

**Additional Complex Manipulation Task with Visual Distractors.** Finally, we implemented a more complex robot manipulation task to further validate RAPL's ability to disentangle both semantic and low-level visual features that underlie an end-user's preferences under visual distractors. The detailed task description and results are reported in Appendix A.7. We find that **RAPL** significantly outperforms all baselines in this more complex manipulation and preference-learning task.

## 6 RESULTS: ZERO-SHOT GENERALIZATION ACROSS EMBODIMENTS

So far, the preference feedback used for aligning the visual representation was given on videos $\mathbf{o} \in \tilde{\Xi}$ generated on the same embodiment as that of the robot. However, in reality, the human could give preference feedback on videos of a *different* embodiment than the specific robot's. We investigate if our approach can *generalize* to changes in the embodiment between the preference dataset $\tilde{S}_\mathrm{H}$ and the robot policy optimization.

**Tasks & Baselines.** We use the same setup for each environment as in Section 5.

**Cross-Domain Agents.** In X-Magical, reward functions are always trained on the *short stick* agent, but the learning agent is a *gripper* in **avoid** and a *medium stick* in **grouping** task. In manipulation we train RAPL and RLHF on videos of the *Franka*, but deploy the rewards on the *Kuka* robot.

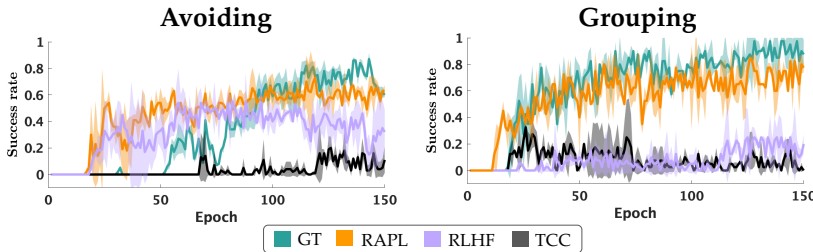

Figure 8: **Cross-Embodiment: X-Magical.** Policy evaluation success rate during policy learning. Colored lines are the mean and variance of the evaluation success rate. RAPL achieves a comparable success rate compared to GT with high learning efficiency, and outperforms RLHF and TCC.

**Hypothesis.** *RAPL enables zero-shot cross-embodiment generalization of the visual reward compared to other visual rewards.*

**Results.** Figure 8 and Figure 9 show the policy evaluation histories during RL training with each reward function in the cross-embodiment X-Magical environment and the manipulation environment. We see that in all cross-embodiment scenarios, **RAPL** achieves a comparable success rate compared to **GT** and significantly outperforms baselines which struggle to achieve more than zero success rate, supporting **H1**. See more results in App. A.6.

We note an interesting finding in the X-Magical grouping task when the representation is trained on videos of the *short stick* agent, but the learning agent is the *medium stick* agent (see Figure 14 in Appendix). Because the *short stick* agent is so small, it has a harder time keeping the objects grouped together; in-domain results from Section 5 show a success rate of ≈ 60% (see Figure 4). In theory, with a well-specified reward, the task success rate should **increase** when the *medium stick* agent does the task, since it is better suited to push objects together. Interestingly, when the *short stick* visual representation is transferred zero-shot to the *medium stick*, we see precisely this: **RAPL**'s task success rate **improves** by 33% under cross-embodiment transfer (succ. rate ≈ 80%). This finding indicates that RAPL can learn task-relevant features that can guide correct task execution even on a new embodiment.

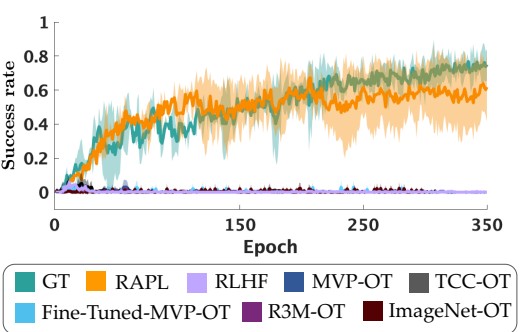

Figure 9: **Cross-Embodiment: Manipulation.** Colored lines are the mean and variance of the success rate during policy learning.

## 7 CONCLUSION

In this work, we presented a video-only, preference-based learning method for solving the visual representation alignment problem. We demonstrated that with an aligned visual representation, reward learning via optimal transport feature matching can generate successful robot behaviors with high sample efficiency, and shows strong zero-shot generalization when the visual reward is learned on a different embodiment than the robot's.

Although in this work we focused on controlled simulation experiments, future work should validate RAPL with real human feedback and robotic hardware experiments. Though our method shows better sample efficiency than RLHF, asking humans for preference queries should be done strategically (e.g., via active learning), should be robust to noisy feedback, and could be improved by leveraging multi-modality (e.g., preferences & language feedback). While our current approach was an offline fine-tuning method, future work onto *online* visual reward fine-tuning from human feedback is an exciting direction. Finally, incorporating feedback from multiple humans (e.g., crowd-sourced multimodal preferences) are also an exciting future direction.

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

## A APPENDIX

### A.1 MOTIVATING QUESTIONS

Inspired by the appendix of (Karamcheti et al., 2023), in this section, we list some motivating questions that may arise from reading the main paper.

***Q1.*** *The experiments all consider preferences beyond task progress. If the end-user's preference is only progress, can RAPL achieve comparable performance compared to the SOTA TCC-based visual reward (Zakka et al., 2022)?*

To investigate this, we return to the X-Magical **grouping** task (middle plots in Figure 2) where a short stick robot needs to push two objects to goal. We removed the grouping preference so the ground truth task reward is consistent with the original benchmark in (Zakka et al., 2022). We trained RAPL with 150 preference queries and compare it with the TCC reward model trained using 500 demonstrations. In Figure 10, we show the policy evaluation success rate during policy learning. We see that **RAPL** has comparable final success rate compared to **TCC** and has a more stable policy training, showing that it can learn a superset of preferences when compared to TCC.

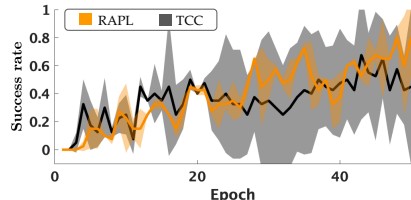

Figure 10: **X-Magical.** Progress-only reward success rate.

***Q2.*** *What makes RAPL different from prior robot learning works that use optimal transport (OT) based visual rewards?*

Indeed, OT-based visual rewards have become increasingly popular for learning robot manipulation (Haldar et al., 2023b;a; Guzey et al., 2023). However, key to making the OT-based visual reward successful in Haldar et al. (2023b) is fine-tuning the representation model via behavior cloning tasks. This helps the model to capture some task-relevant information at the cost of requiring action labels. Furthermore, by relying on action labels, it is unclear if the learned reward can generalize to a different embodiment. Instead, our approach learns the representation only using preference queries (no action labels) and can generalize to embodiments.

***Q3.*** *Why do **MVP-OT** and TCC-OT achieve near 0 success rate in the robot manipulation experiments in Figure 6 and Figure 9?*

Recall that both **MVP-OT** and **TCC-OT** use optimal transport to match the embedding distribution of the robot and the expert, but they vary which visual representation they use to obtain the embedding.

The MVP encoder is trained via masked autoencoding (He et al., 2022) to reconstruct heavily masked video frames. As such, it captures representations amenable to per-pixel reconstruction. Prior work (Karamcheti et al., 2023) has demonstrated that this representation struggles with higher-level problems (e.g., language-based imitation). We hypothesize this is why **MVP-OT** struggles to capture preference-relevant features and does not lead to aligned robot behaviors. Our results are also consistent with the experiments in (Haldar et al., 2023b) where an OT-based visual reward with a pre-trained MVP representation model gives near 0 success rate for manipulation tasks.

The TCC encoder is trained via temporal cycle-consistency constraints, and as such captures representations that encode solely task progress (e.g., distance to the goal image). Such a representation works well when goal reaching is the only preference of the end user. In our tabletop grouping task, the end user cares about goal reaching, but they also prefer moving the two objects together to goal region over moving the objects one-by-one. Thus if the robot happens to push one object towards the goal during policy learning, **TCC-OT** will reward this behavior (since this image is getting "closer" to the goal image) even though this is not preferred by the user.

### A.2 EXTENDED RELATED WORK

**Visual robot rewards** promise to capture task preferences directly from videos. Self-supervised approaches leverage task progress inherent in video demonstrations to learn how "far" the robot

is from completing the task (Zakka et al., 2022; Kumar et al., 2023; Ma et al., 2023) while other approaches identify task segments and measure distance to these subgoals (Sermanet et al., 2016; Tanwani et al., 2020; Shao et al., 2020; Chen et al., 2021). However, these approaches fail to model preferences *during* task execution that go beyond progress (e.g., spatial regions to avoid during movement). Fundamental work in IRL uses feature matching between the expert and the learner in terms of the expected state visitation distribution to infer rewards (Abbeel & Ng, 2004; Ziebart et al., 2008), and recent work in optimal transport has shown how to scale this matching to high dimensional state spaces (Xiao et al., 2019; Dadashi et al., 2021; Papagiannis & Li, 2022; Luo et al., 2023). However, key to making this matching work from high-dimensional visual input spaces is a good visual embedding. Previous works used proxy tasks, such as behavior cloning (Haldar et al., 2023a;b) or temporal cycle-consistency learning (Dadashi et al., 2021), to train the robot's visual representation. In contrast to prior works that rely on hard-to-obtain action labels or using only self-supervised signal, we propose an OT-based visual reward that is trained purely on videos (no action labels needed) that ranked by the *end-user's* preferences.

**Preference-based learning.** While demonstrations have been the data of choice for reward learning in the past, an increasingly popular approach is to use preference-based learning (Christiano et al., 2017; Sadigh et al., 2017; Biyik & Sadigh, 2018; Wirth et al., 2017; Brown et al., 2019; Stiennon et al., 2020; Zhang et al., 2022; Shin et al., 2023). Here the human is asked to compare two (or more) trajectories (or states), and then the robot infers a map from ranked trajectories to a scalar reward. This feedback is often easier for people to give than kinesthetic teaching or fine-grained feedback (Shin et al., 2023). At the same time, prior works and our experiments show that directly predicting the reward from preference queries and high-dimensional input suffers from high sample inefficiency and causal confusion (Bobu et al., 2023b; Tien et al., 2022). To mitigate this issue, (Brown et al., 2020) augments multiple self-supervised objectives like inverse dynamics prediction or enforcing temporal cycle-consistency with the preference learning loss, but this requires additional signals like actions and the additional self-supervised objective may bias the learned rewards towards capturing spurious correlations.

**Representation alignment in robot learning.** Representation alignment studies the agreement between the representations of two learning agents. As robots will ultimately operate in service of people, representation alignment is becoming increasingly important for robots to interpret the world in the same way as we do. Previous work has leveraged user feedback, such as human-driven feature selection (Bullard et al., 2018; Luu-Duc & Miura, 2019), interactive feature construction (Bobu et al., 2021; Katz et al., 2021), or similarity-implicit representation learning (Bobu et al., 2023a), to learn aligned representations for robot behavior learning. But they either operate on a manually defined feature set or learning features in low-dimensional state space settings (e.g., positions). In the visual domain, (Zhang et al., 2020) uses a per-image reward signal to align the image representation with the preferences encoded in the reward signal; however, when the main objective is learning the human's reward then assuming a priori access to such a reward signal is not feasible. Instead, our work utilizes human preference feedback to align the robot's visual representations with the end user and optimal transport as our embedding-based reward function.

## A.3 Optimal Transport Based Reward

**Setup.** Let $\mathbf{o} = \{o^t\}_{t=1}^{t=T}$ be a trajectory of observations, where $T$ is the trajectory length. Let $\mathcal{D}_+ \subset \mathcal{S}_{\phi_H}$ be a dataset of preferred videos from the preference video dataset and $\mathcal{D}_R$ be the set of videos induced by a given robot policy $\pi_R$. We denote $\phi : \mathbb{R}^{h,w,3} \to \mathbb{R}^{n_e}$ as an observation encoder that maps a $h \times w$ RGB image to a $n_e$ dimensional embedding. For any video $\mathbf{o}$, let the induced empirical embedding distribution be $\rho = \frac{1}{T} \sum_{t=0}^{T} \delta_{\phi_R(o^t)}$, where $\delta_{\phi_R(o^t)}$ is a Dirac distribution centered on $\phi_R(o^t)$.

**Background.** Optimal transport finds the optimal coupling $\mu^* \in \mathbb{R}^{T \times T}$ that transports the robot embedding distribution, $\rho_R$, of a robot video $\mathbf{o}_R \in \mathcal{D}_R$ to the expert video embedding distribution, $\rho_+$ for $\mathbf{o}_+ \in \mathcal{D}_+$, with minimal cost (as measured by a distance function, e.g. cosine distance). This comes down to an optimization problem that minimizes the Wasserstein distance between the two distributions:

$$\mu^* = \operatorname*{arg\,min}_{\mu \in \mathcal{M}(\rho_R, \rho_+)} \sum_{t=1}^{T} \sum_{t'=1}^{T} c\big(\phi(o_R^t), \phi(o_+^{t'})\big) \mu_{t,t'}. \tag{7}$$

where $\mathcal{M}(\rho_\mathrm{R}, \rho_+) = \{\mu \in \mathbb{R}^{T \times T} : \mu\mathbf{1} = \rho_\mathrm{R}, \mu^T\mathbf{1} = \rho_+\}$ is the set of coupling matrices and $c : \mathbb{R}^{n_\mathrm{R}} \times \mathbb{R}^{n_+} \to \mathbb{R}$ is a cost function defined in the embedding space (e.g., cosine distance). The optimal transport plan gives rise to the following reward signal that incentivizes the robot to stay within the expert demonstration distribution by explicitly minimizing the distance between the observation distribution and expert distribution:

$$r(o_\mathrm{R}^t; \phi_\mathrm{R}) = -\sum_{t'=1}^{T} c\big(\phi_\mathrm{R}(o_\mathrm{R}^t), \phi_\mathrm{R}(o_+^{t'})\big)\mu_{t,t'}^*. \tag{8}$$

**Regularized optimal transport.** Solving the above optimization in Equation 7 exactly is generally intractable for high dimensional distributions. In practice, we solve a entropy regularized version of the problem following the Sinkhorn algorithm (Peyré et al., 2019) which is amenable to fast optimization:

$$\mu^* = \underset{\mu \in \mathcal{M}(\rho_\mathrm{R}, \rho_+)}{\arg\min} \sum_{t=1}^{T} \sum_{t'=1}^{T} c\big(\phi(o_\mathrm{R}^t), \phi(o_+^t)\big)\mu_{t,t'} - \epsilon\mathcal{H}(\mu), \tag{9}$$

where $\mathcal{H}$ denotes the entropy term that regularizes the optimization and $\epsilon$ is the associated weight.

**Choosing an $\mathbf{o}_+$ to match with $\pi_\mathrm{R}$.** The reward (8) requires matching the robot to an expert observation video. To choose this expert observation, we follow the approach from (Haldar et al., 2023a). During policy optimization, given a robot's trajectory's observation $\mathbf{o}_\mathrm{R}$ induced by the robot policy $\pi_\mathrm{R}$, we select the the "closest" expert demonstration $\mathbf{o}_+^* \in \mathcal{D}_+$ to match the robot behavior with. This demonstration selection happens via:

$$\mathbf{o}_+^* = \underset{\mathbf{o}_+ \in \mathcal{D}_+}{\arg\min} \ \underset{\mu \in \mathcal{M}(\rho_\mathrm{R}, \rho_+)}{\min} \sum_{t=1}^{T} \sum_{t'=1}^{T} c\big(\phi(o_\mathrm{R}^t), \phi(o_+^{t'})\big)\mu_{t,t'}. \tag{10}$$

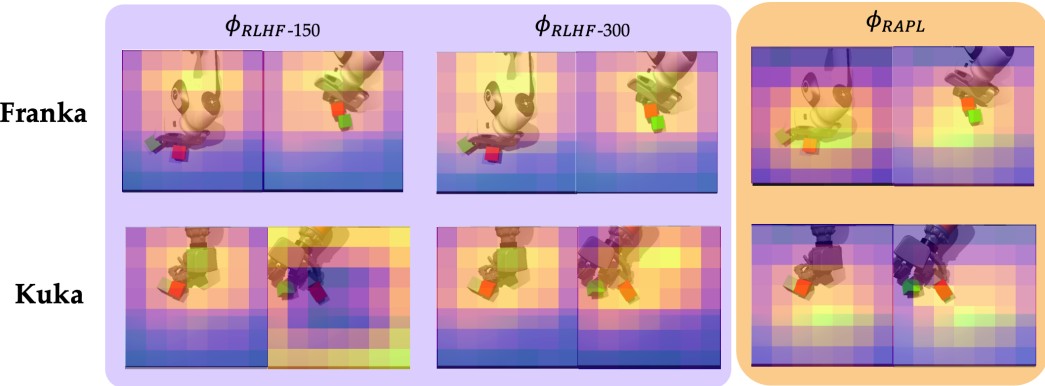

Figure 11: **Manipulation: Attention Map.** Visualization of attention map for RLHF-150 demos, RLHF-300 demos, and RAPL with 150 demos for both Franka and Kuka (cross-embodiment) images. Each entry of the figure shows two image snapshots from the relevant demonstration set with the attention map overlaid. Bright yellow areas indicate image patches that contribute most to the final embedding; darker purple patches indicate less contribution. $\phi_{RLHF-150}$ is biased towards paying attention to irrelevant areas that can induce spurious correlations; in contrast RAPL learns to focus on the task-relevant objects and the goal region. $\phi_{RLHF-300}$'s attention is slightly shifted to objects while still pays high attention to the robot embodiment.

## A.4 ATTENTION MAP FOR RAPL AND RLHF

In Figure 11 we visualize the attention map with a novel use of *linear permutation* and *kernel inflation* (Xu et al., 2022). Specifically, we use channel-averaged 2D feature map (*i.e.,* activation map) as our attention map (Xu et al., 2020). Different from previous works that operate 2D feature maps, our approach utilizes a linear mapping $\mathcal{W} \in \mathcal{R}^{C_{in} \times C_{out}}$ on a 1D feature, which is average-pooled from a 2D feature $\mathcal{F}_{2D} \in \mathcal{R}^{C_{in} \times H \times W}$. Mathematically, the procedure can be formulated

as

$$\hat{\mathcal{F}}_{1D} = \frac{\mathcal{W}^T \times \sum_i^{H \times W} \mathcal{F}_{2D}^i}{H \times W} \tag{11}$$

where $\hat{\mathcal{F}}_{1D} \in \mathcal{R}^{C_{out} \times 1}$ is the aligned features from our proposed RAPL by $\mathcal{W} \in \mathcal{R}^{C_{in} \times C_{out}}$. Inspired by Xu et al. (2022), we can inflate the 1D linear mapping $\mathcal{W} \in \mathcal{R}^{C_{in} \times C_{out}}$ into 2D and keep the kernel size as 1, *i.e.*, $\mathcal{W} \in \mathcal{R}^{C_{in} \times C_{out}} \rightarrow \mathcal{W}_{inflate} \in \mathcal{R}^{C_{in} \times C_{out} \times 1 \times 1}$. Then above equation can be equally represented as

$$\hat{\mathcal{F}}_{2D} = \mathcal{W}_{inflate}^T \times \mathcal{F}_{2D}, \hat{\mathcal{F}}_{1D} = \frac{\sum_i^{H \times W} \hat{\mathcal{F}}_{2D}^i}{H \times W} \tag{12}$$

We average the $\hat{\mathcal{F}}_{2D} \in \mathcal{R}^{C_{out} \times H \times W}$ in channel dimension, and visualize the output as our attention map. A visualization of the full process is shown in Figure 12.

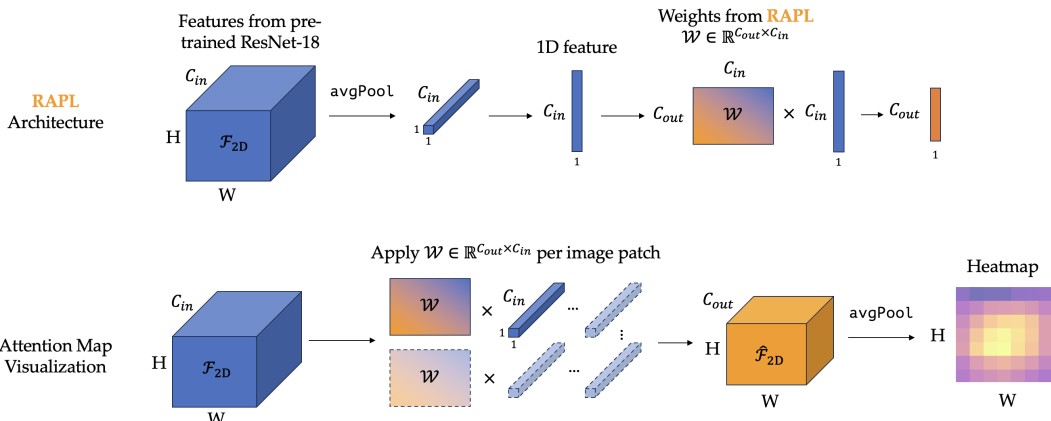

Figure 12: **Attention Map Visualization Method.** (top) Visualization of our RAPL architecture. (bottom) Visualization of our process for creating a 2D attention map.

## A.5 ADDITIONAL RLHF RESULTS: ABLATION ON FEEDBACK DATASET SIZE

In Section 5.2, it's surprising that RLHF fails to learn a robot policy in a more realistic environment since its objective is similar to ours, but without explicitly considering representation alignment. To further investigate this, we apply a linear probe on the final embedding and visualize the image heatmap of what RAPL's (our representation model trained with 150 training samples), RLHF-150's (RLHF trained with 150 samples), and RLHF-300's (RLHF trained with 300 samples samples) final embedding pays attention to in Figure 11.

We see that $\phi_{RAPL}$ learns to focus on the objects, the contact region, and the goal region while paying less attention to the robot arm; $\phi_{RLHF-150}$ is biased towards paying attention to irrelevant areas that can induce spurious correlations (such as the robot arm and background area); $\phi_{RLHF-300}$'s attention is slightly shifted to objects while still pays high attention to the robot embodiment.

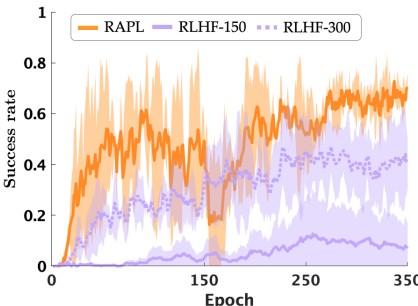

Figure 13: **Manipulation.** RAPL outperforms RLHF by 75% with 50% less training data.

When deploying $\phi_{RLHF-300}$ in Franka manipulation policy learning, we observe that policy performance is slightly improved (indicating that with more feedback data, preference-based reward prediction could yield to an aligned policy), but RAPL still outperforms RLHF by 75% with 50% less training data, supporting the hypothesis: *RAPL outperforms RLHF with lower amounts of human preference queries.*

## A.6 ADDITIONAL CROSS-EMBODIMENT RESULTS: X-MAGICAL & KUKA MANIPULATION

Figure 14 shows the rewards over time for the three cross-embodiment video observations (marked as preferred by the end-user's ground-truth reward or disliked) in the avoid (left) and group task (right). Across all examples, RAPL 's rewards are highly correlated with the GT rewards even when deployed on a cross-embodiment robot.

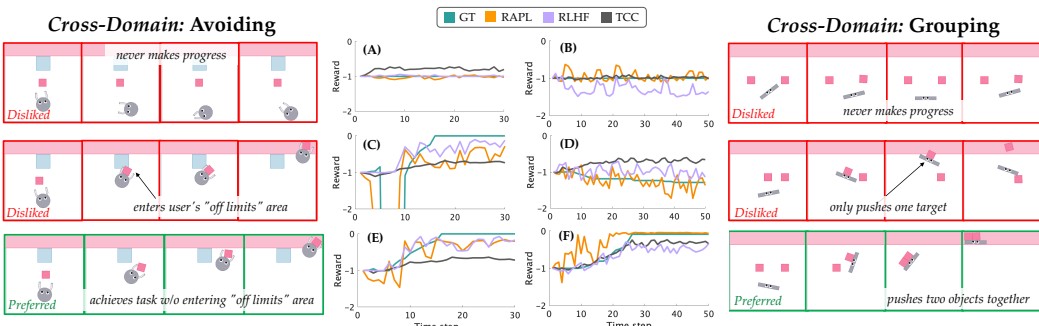

Figure 14: **Cross-Embodiment: X-Magical.** RAPL can distinguish preferred and disliked videos in the cross-embodiment setting.

Figure 15 shows the rewards over time for the two cross-embodiment video observations (marked as preferred by the end- user's ground-truth reward or disliked). Across all examples, RAPL 's rewards are highly correlated with the GT rewards even when deployed on a cross-embodiment robot.

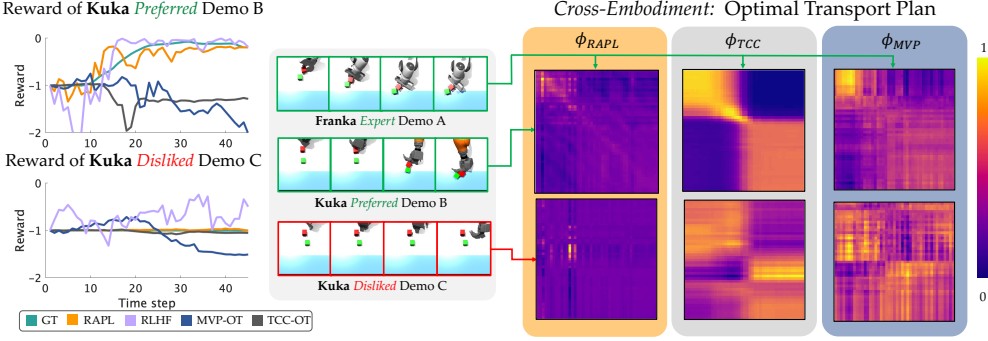

Figure 15: **Cross-Embodiment: Manipulation.** (center) Expert video on *Franka* robot, preferred video on *Kuka*, and disliked *Kuka* video demo. (left) Reward associated with each video under each method. RAPL's predicted reward generalizes to the *Kuka* robot and follows the GT pattern. (right) OT plan for each representation. Columns are embedded frames of expert demo on *Franka* robot. Rows of top matrices are embedded frames of preferred demo on *Kuka*; rows of bottom matrices are embedded frames of disliked demo on *Kuka*. Peaks exactly along the diagonal indicate that the frames of the two videos are aligned in the latent space; uniform values in the matrix indicate that the two videos cannot be aligned (i.e., all frames are equally "similar" to the next). RAPL matches this structure: diagonal peaks for expert-and-preferred and uniform for expert-and-disliked, while baselines show diffused values no matter the videos being compared.

## A.7 ADDITIONAL COMPLEX ROBOT MANIPULATION TASK WITH VISUAL DISTRACTORS

In this section, we consider a more complex robot manipulation task to further validate RAPL's ability to disentangle visual features that underlie an end-user's preferences. We increase the difficulty of the the robot manipulation environment described in subsection 5.2 by adding visual distractors that are irrelevant to the human's preferences (left figure in Figure 16). The environment has

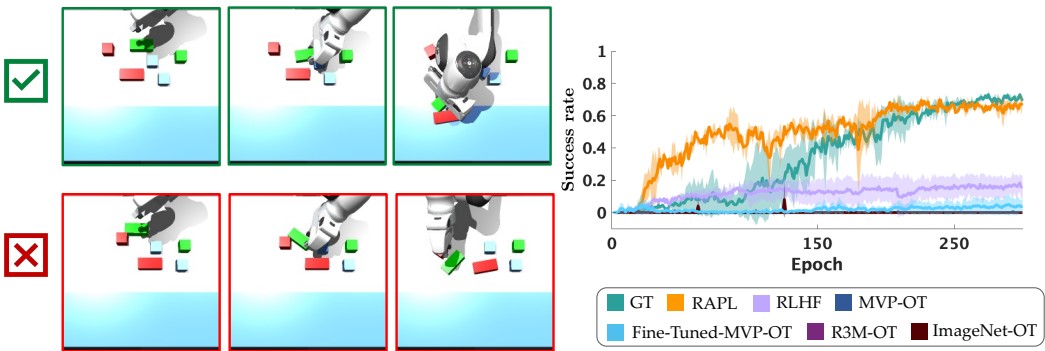

Figure 16: **Clutter task: Complex robot manipulation with visual distractors**. (left) Visualization of preferences. Green (preferred behavior): the Franka arm grasps the green rectangular prism and uses that to push the red rectangular prism to the goal region. Red (negative behavior): the Franka arm fails to grasp the rectangular prism. (right) Policy training results for our method and additional baselines.

multiple objects on the table of various colors—red, green, and goal-region-blue—and some of the objects are cubes while others are rectangular prisms. The Franka robot arm needs to learn that the end-user prefers to push the *rectangular* objects (instead of the cubes) *efficiently together* (instead of one-at-a-time) to the goal region.

**Task Complexity & Feature Entanglement.** Compared to the manipulation task described in subsection 5.2, the Franka arm needs to learn representations that can disentangle both semantic preference (grouping) and low-level preference (shape) from RGB images under visual distractors (object color). In addition, the Franka arm needs to learn to *grasp* the rectangular prism first and use that to push the second rectangular prism, as it is difficult to directly push the rectangular prism stably using the robot finger gripper, thus increasing the task difficulty.

**Privileged State & Reward.** The state $s$ is 34D: robot proprioception ($\theta_{\text{joints}} \in \mathbb{R}^{10}$), 3D object positions ($p_{\text{obj}_{\text{rect}}^{1,2}}, p_{\text{obj}_{\text{cube}}^{1,\ldots,4}}$), and object distances to goal ($d_{\text{goal2obj}_{\text{rect}}^{1,2}}, d_{\text{goal2obj}_{\text{cube}}^{1,\ldots,4}}$). The simulated human's reward is:

$$r_{\text{group}}^*(s) = -\max(d_{\text{goal2obj}^{\text{rect},1}}, d_{\text{goal2obj}^{\text{rect},2}}) - ||p_{\text{obj}_{\text{rect}}^1} - p_{\text{obj}_{\text{rect}}^2}||_2$$
$$- 0.1 \sum_{i=1}^{4} ||p_{\text{obj}_{\text{cube}}^i} - p_{\text{obj}_{\text{cube}}^{i,\text{init}}}||_2.$$

**Baselines.** In addition to comparing **RAPL** against (1) **GT** and (2) **RLHF**, we ablate the *representation model* but control the *visual reward structure*. We consider four additional baselines that all use an optimal transport-based reward but operate on different representations: (3): **MVP-OT** which is a off-the-shelf visual transformer (Xiao et al., 2022) pre-trained on the Ego4D data set (Grauman et al., 2022) via masked visual pre-training; (4): **Fine-Tuned-MVP-OT** which fine-tunes **MVP-OT** using images from the task environment via LoRA (Hu et al., 2021); (5): **R3M-OT** which is an off-the-shelf ResNet-18 encoder (Nair et al., 2022) pre-trained on the Ego4D data set (Grauman et al., 2022) via a learning objective that combines time contrastive learning, video-language alignment, and a sparsity penalty; (6): **ImageNet-OT** which is an off-the-shelf ResNet-18 encoder pre-trined on ImageNet. We use the same preference dataset with 150 triplets for training RLHF and RAPL.

**Results.** The right plot in Figure 16 shows the policy evaluation history during RL training with each reward function. We see **RAPL** performs comparably to **GT** (succ. rate: $\approx 65\%$) while all baselines struggle to achieve a success rate of more than $\approx 10\%$ with the same number of epochs.

## A.8 SPEARMAN'S CORRELATION BETWEEN GT AND LEARNED VISUAL REWARDS

In this section, we conduct a quantitative analysis to investigate the relationship between the learned visual reward and the end-user's ground-truth reward. Specifically, for each robot manipulation task, we compute the average Spearman's correlation coefficient between the learned visual reward

and the end-user's ground-truth reward across 100 video trajectories. We can observe from Table 1 that our learned visual reward shows a stronger correlation to the end-user's ground-truth reward compared to baselines.

| | Spearman's Correlation | | |
| --- | --- | --- | --- |
| | **Franka Group** (Sec. 5.2) | **Kuka Group** (Sec. A.6) | **Franka Clutter** (Sec. A.7) |
| **RAPL** | 0.59 | 0.47 | 0.61 |
| **RLHF** | 0.38 | 0.31 | 0.26 |
| **MVP-OT** | -0.1 | 0.02 | 0.08 |
| **Fine-Tuned-MVP-OT** | 0.19 | 0.02 | 0.11 |
| **ImageNet-OT** | -0.09 | 0.12 | -0.02 |
| **R3M-OT** | 0.03 | -0.14 | -0.17 |

Table 1: Spearman's rank correlation coefficient between the **GT** reward and each learned reward.

## A.9 ROBOT MANIPULATION: RLHF PERCEIVED VS. TRUE SUCCESS

We investigated if the poor RLHF performance in Figure 6 can be attributed to poor RL optimization or to poor visual reward structure. We compared the true success rate of the RLHF policy (under the true human's measure of success) to the "perceived" performance under the RLHF reward. These results are visualized in Figure 17: purple is the true success rate and black is the "perceived" reward under the RLHF learned reward. We see that after 350 epochs, the RLHF learned reward perceives the policy as achieving a high reward. However, as shown in the manuscript's Figure 6 and in Figure 17, the true success rate is still near zero. This indicates that the RL optimization is capable of improving over time, but it is optimizing a poor reward signal that does not correlate with the true measure of success.

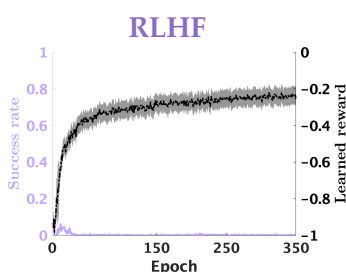

Figure 17: **Manipulation.** True success rate of the RLHF policy (purple) versus the "perceived" performance under the RLHF reward (black).

