# OpenReview forum: "What Matters to You? Towards Visual Representation Alignment for Robot Learning"
_ICLR.cc/2024/Conference — ICLR 2024 poster_

### Official Review · Reviewer_2gN6 · 2023-10-30

**Soundness:** 2 fair
**Presentation:** 2 fair
**Contribution:** 2 fair
**Rating:** 6
**Confidence:** 3

**Summary:**

The paper attempts to utilize human feedback to synchronize visual representation with the end-user's priorities for the task at hand. The authors present the Representation-Aligned Preference-based Learning (RAPL) method to tackle challenges in visual representation alignment and visual reward learning. Trials conducted in XMAGICAL and robotic manipulation demonstrate that RAPL consistently produces desired robot behaviors with notable sample efficiency and robust zero-shot generalization.

**Strengths:**

The paper pioneers the formalization of the visual representation alignment challenge in robotics, framing it as metric learning within the human representation space. Their proposed RAPL targets resolving the alignment issue and facilitates learning visual robot rewards through optimal transport. Experimental results further demonstrate that RAPL consistently yields desired robot behaviors, maintaining high sample efficiency. Lastly, they also show zero-shot generalization across embodiments.

**Weaknesses:**

1. The task complexity may not adequately highlight the significance of human feedback in robot learning.
2. While the approach captures human-like preferences, there's no concrete evidence to validate its representation of genuine human preferences in real-world tasks. Enhancing the complexity of preference-based feedback might better align pre-trained visual models with user preferences.
3. The author should elaborate in the main paper on their method of utilizing privileged access to state information for dataset generation.
4. While the efficacy of this approach is demonstrated with the X-Magical and ISACCGym robots, it would be beneficial to see its applicability in other robotic embodiment or real-world scenarios.

**Questions:**

All listed in the weakness section. I would consider changing my rating, if the author could address my questions.

---

> ### Author Response · Authors · 2023-11-22
> **Responses to reviewer 2gN6**
>
> We would like to thank the reviewer for taking the time to review our paper and provide valuable feedback. We are excited to have the chance to address the reviewer’s questions and concerns. These edits will make the paper stronger.
>
> ***Q1.  The task complexity may not adequately highlight the significance of human feedback in robot learning.***
>
> A1. We would like to thank the reviewer for the suggestion. Per your suggestion, we implemented a more complex robot manipulation task (described in detail in the common response and in the Appendix of the updated manuscript). Specifically, we modified the robot manipulation environment described in subsection 5.2 by 1) increasing the manipulation difficulty of the task (more objects of various colors and sizes, robot needs to grasp the object while pushing), 2) introducing a more complex preference (grouping by object size), and 3) adding visual distractors (object color). Please find the detailed description in Appendix Section A.8. We find that in this more complex manipulation and preference-learning task, our method significantly outperforms all the baselines  (including the new baselines as introduced in our common response).
>
>
> ***Q2. While the approach captures human-like preferences, there's no concrete evidence to validate its representation of genuine human preferences in real-world tasks. Enhancing the complexity of preference-based feedback might better align pre-trained visual models with user preferences.***
>
> A2. In this paper, we focused on learning representations and visual rewards from simulated human feedback, which indeed could be different from real human preference feedback. We agree that learning from real end-user feedback is extremely interesting. However, we focused on simulated human feedback in this investigation for several reasons. First, using simulated human feedback allows us to carefully ablate the size of the preference dataset. Second, it gives us privileged access to the ground truth task reward for direct comparison, which ultimately enables us to more rigorously evaluate our idea. Third, evaluating our method with real user feedback requires the Institutional Review Boards (IRB) approval, which is time consuming and requires detailed evaluation results on simulated humans. Based on your suggestion, we are currently preparing our IRB application given the promising results demonstrated in the paper and we are excited to evaluate our method with real user feedback through IRB approved user studies in our future work.
>
> ***Q3. How does our paper utilizes the privileged access to state information for dataset generation?***
>
> A3.  The procedure of generating the preference dataset is described in the second paragraph of Section 4. In summary, we utilize privileged state information of the environment and a designed reward function to rank trajectories for constructing the preference dataset. The detailed information about the privileged state and reward function for the X-magical environment is described in second paragraph of Section 5.1, and the detailed information about the privileged state and reward function for the manipulation environment is described in third paragraph of Section 5.2.
>
> ***Q4. While the efficacy of this approach is demonstrated with the X-Magical and ISACCGym robots, it would be beneficial to see its applicability in other robotic embodiment or real-world scenarios.***
>
> A.4 We would like to thank the reviewer for the helpful suggestion. We agree that deploying this approach in more robotic embodiments and real-world scenarios is very interesting. The main goal of this paper is to demonstrate that with an aligned visual representation, reward learning via optimal transport feature matching can generate successful robot behaviors with high sample efficiency, and show strong zero-shot embodiment generalization. Since our work is the first step towards visual representation alignment for robot learning, we focused on robotic experiments in simulation to thoroughly investigate our approach and control for confounds. We highlight in the conclusion that demonstrating RAPL with real human videos, preferences, and robotic hardware experiments is important future work.

---

> > ### Comment · Reviewer_2gN6 · 2023-11-22
> > **Reponse to the author**
> >
> > I want to thank the author for the response. After much consideration, I have decided to raise my rating.

---

> > > ### Author Response · Authors · 2023-11-22
> > >
> > > Dear reviewer 2gN6,
> > >
> > > We are glad that our responses could address your questions, and we appreciate your reconsideration of the score of our paper! Please let us know if you have further questions. Thank you again for taking the time to review our paper and providing insightful comments!
> > >
> > > Best regards,
> > >
> > > Authors

---

### Official Review · Reviewer_vWH8 · 2023-10-31

**Soundness:** 3 good
**Presentation:** 4 excellent
**Contribution:** 2 fair
**Rating:** 3
**Confidence:** 3

**Summary:**

This paper proposes Representation-Aligned Preference-based Learning (RAPL), a method which leverages human feedback to align visual representations with the end-user. Preference-based learning and optimal transport are used for this alignment and for the policy learning. Experimental results in two different environments reveal that RAPL outperforms prior work by learning a policy which both leads to higher rewards and has better sample efficiency.

**Strengths:**

- Well-written. The paper is very well-written and easy to follow. The figures very much aid in understanding the paper.
- Experiments. The experiments reveal that the proposed methodology outperforms baseline methods by non-trivial margins. The experiments are done across a number of environments, and evaluate the proposed methodology in a number of different ways.
- Ablations and Visualizations. Ablations are conducted and visualizations are included to better understand how and why the proposed methodology performs better than the baselines.

**Weaknesses:**

- Missing / weak baselines.
  - Zhang et al (2020) is described to be the most similar to the proposed methodology. Why is this not compared to?
  - The main baseline appears to be TCC (Zakka et al., 2022; Kumar et al., 2023). Which of these (Zakka et al., 2022 vs Kumar et al., 2023) is used as the baseline? Furthermore, both Zakka et al., 2022 and Kumar et al., 2023 report results on {long-stick, medium-stick, short-stick, gripper} tasks -- why are these tasks not used to report results, for a completely fair comparison? This would be the most revealing comparison to the baselines.
  - A comparison to other pre-training work is made by comparing to MVP-OT, which is revealing. MVP is trained on Ego4D, so there is a massive distribution shift, perhaps explaining the poor performance. For a fairer comparison, could Masked Visual Pre-training be performed on data from the environments directly (e.g. using the trajectories)?
  - While the visual backbone models are initialized with ImageNet pre-trained weights, may be revealing to include this (without any further training) as another baseline.
- Data is simulated. A simulated human model is used, rather than learning from real end-user feedback.
- Figure 3 qualitatively shows the reward correlation -- quantification of such correlation using e.g. Spearman's correlation coefficient, compared to baselines, would be revealing.

**Questions:**

- How is the proposed approach better than GT for "Avoiding"?
- There seem to be error bars on many of the plots -- how many trials is this over?

---

> ### Author Response · Authors · 2023-11-22
> **Responses to reviewer vWH8 [1/2]**
>
> We would like to thank the reviewer for taking the time to review our paper and provide valuable feedback. We are glad that the reviewer thinks our paper is sound and well presented. We are excited to have the chance to address the reviewer’s questions and concerns. These edits will make the paper stronger.
>
> ***Q1.  Missing / weak baselines.***
>
> ***Q1.1 Why not compare with Zhang et al (2020).***
>
> A1.1 Indeed, this paper is related to our work as it studies representation alignment through reward signals. Specifically, this paper uses reward signals to learn representations which disentangle high versus low reward images and demonstrate that the learned representations outperformed the representations learned through self-supervised learning. However, a core assumption of the approach is a priori access to a task’s correctly specified reward function. Obtaining a correct reward function is precisely the learning goal of our setting: in other words, the learning agent does not know the human’s reward but needs to learn it through our proposed RAPL approach. Thus, Zhang et al (2020) is not a comparable baseline for our approach. We have modified our wording in Section 1 to make it more clear.
>
> ***Q1.2 Which is our TCC baseline in the X-magical? Zakka et al., 2022 or Kumar et al., 2023. Both Zakka et al., 2022 and Kumar et al., 2023 report results on {long-stick, medium-stick, short-stick, gripper} tasks -- why are these tasks not used to report results, for a completely fair comparison?***
>
> A1.2 The TCC representation baseline in the X-magical environment is from Zakka et al., 2022. Regarding embodiment combinations, we agree that it would be compelling to see all combinations of training and deployment embodiments. In our manuscript, we leveraged all four embodiments, but due to time and space constraints, we only investigated a subset of training and deployment embodiment pairs. Specifically, we chose embodiment pairs based on the hypotheses we were investigating in the paper.  For example, in the cross-embodiment experiment in the X-magical groping task (Section 6), we chose the short-stick as the training embodiment and chose the medium stick as the deployment embodiment to demonstrate that our approach can learn task-relevant representations:  because the short stick agent is so small, it has a harder time keeping the objects grouped together; in-domain results from Section 5 show a success rate of ≈ 60% (see Figure 4). In theory, with a well-specified reward, the task success rate should increase when the medium stick agent does the task, since it is better suited to push objects together. Interestingly, when the short stick visual representation is transferred zero-shot to the medium stick, we see precisely this: RAPL’s task success rate improves by 33% under cross-embodiment transfer (succ. rate ≈ 80%). This finding indicates that RAPL can learn task-relevant features that can guide correct task execution even on a new embodiment.
>
>
> ***Q1.3 Fine-tune the MVP encoder on data from the environments directly and then use the fine-tuned MVP encoder + OT as the baseline.***
>
> A1.3 We would like to thank the reviewer for the baseline suggestion, and we implemented their idea.  In order to preserve the original representations of MVP while adapting to the task domain, we utilized the low rank adaptation technique (LoRA; Hu et al., 2021) to fine-tune the MVP encoder (VIT as the backbone) on the images from the manipulation environment.. We then utilized the fine-tuned MVP representation to generate the OT-based visual reward. The new results are presented in Section A.7 of the appendix. While fine-tuning does slightly improve the MVP performance, this representation still struggles to achieve > 5% success rate, and our proposed method still significantly outperforms this new baseline (Fine-Tuned-MVP-OT) by approximately 70%. We hypothesize that this is due to the signal used during representation learning: RAPL specifically learns representations that disentangle preferred versus disliked robot behavior videos, while MVP learns to in-paint masked image patches (which does not necessarily provide insight into task-relevant visual features).
>
> ***Q1.4 Baseline that directly uses ImageNet pre-trained representation to generate OT-based reward.***
>
> A1.4 We would like to thank the reviewer for the baseline suggestion. We added this as an additional baseline in the manipulation setting. The new results are attached in Section A.7 of the appendix, and we find that our proposed method still significantly outperforms this new baseline (ImageNet-OT) by approximately 70%.

---

> > ### Author Response · Authors · 2023-11-22
> > **Responses to reviewer vWH8 [2/2]**
> >
> > ***Q2.  Data is simulated. A simulated human model is used, rather than learning from real end-user feedback.***
> >
> > A2. We agree that learning from real end-user feedback is extremely interesting. However, we focused on simulated human feedback in this investigation for several reasons. First, using simulated human feedback allows us to carefully ablate the size of the preference dataset. Second, it gives us privileged access to the ground truth task reward for direct comparison, which ultimately enables us to more rigorously evaluate our idea. Third, evaluating our method with real user feedback requires the Institutional Review Boards (IRB) approval, which is time consuming and requires detailed evaluation results on simulated humans. Based on your suggestion, we are currently preparing our IRB application given the promising results demonstrated in the paper and we are excited to evaluate our method with real user feedback through IRB approved user studies in our future work.
> >
> > ***Q3. Quantification of relationship between the GT reward and the learned visual reward.***
> >
> > A3. We would like to thank the reviewer for this helpful suggestion. For each robot manipulation task, we computed the average Spearman’s correlation coefficient across 100 video trajectories, we found that our learned visual reward shows a stronger correlation to the GT reward compared to baselines. We added the detailed results in Section A.9
> >
> > |              | Franka Group (Sec. 5.2) | Kuka Group (Sec. A.6) | Franka Clutter (Sec. A.8) |
> > |--------------|-------------------------|-----------------------|---------------------------|
> > | **RAPL**     | 0.59                    | 0.47                  | 0.61                      |
> > | **RLHF**     | 0.38                    | 0.31                  | 0.26                      |
> > | **MVP-OT**   | -0.1                    | 0.02                  | 0.08                      |
> > | **Fine-Tuned-MVP-OT** | 0.19            | 0.02                  | 0.11                      |
> > | **ImageNet-OT** | -0.09                | 0.12                  | -0.02                     |
> > | **R3M-OT**   | 0.03                    | -0.14                 | -0.17                     |
> >
> > Table 1: Spearman’s rank correlation coefficient between the GT reward and each learned reward.
> >
> >
> >
> > ***Q4. How is the proposed approach better than GT for "Avoiding"?***
> >
> > A4. Our hypothesis is that the manually-defined GT reward (without extensive reward weight tuning) is not “shaped” as well as the preference-based reward to enable fast RL policy optimization. We would like to stress that the manually-defined GT reward *does* capture all the right features and their relative importance, but from a RL policy learning perspective, we hypothesize that human preferences actually help shape the visual reward in such a way that makes RL optimization easier compared to using the manually-defined reward. However, we would like to note that both the GT reward policy and our visual reward policy eventually converge to a similar final success rate.
> >
> >
> > ***Q5. There seem to be error bars on many of the plots -- how many trials is this over?***
> >
> > A5. We would like to thank the reviewer for reminding us about this missing information. The success rate plots show the mean and standard deviation of five trials with different random seeds. We added this information to the main paper.

---

> > > ### Author Response · Authors · 2023-11-22
> > >
> > > Dear reviewer vWH8,
> > >
> > > We would like to express our appreciation for your valuable feedback and suggestions!
> > >
> > > We are wondering if our additional experiment results on the suggested baselines, along with the new quantitative analysis of the quality of the learned visual rewards, and answers to other questions, address your comments about the paper?
> > >
> > > If you have any further questions, we would be happy to provide additional information!
> > >
> > > Best regards,
> > >
> > > Authors

---

### Official Review · Reviewer_sCYJ · 2023-11-01

**Soundness:** 3 good
**Presentation:** 3 good
**Contribution:** 2 fair
**Rating:** 6
**Confidence:** 4

**Summary:**

The paper proposes a new method called Representation-Aligned Preference-based Learning (RAPL) to align robot’s visual representation using human preferences. The paper formally defines visual representation alignment problem for robotics as minimizing divergence between human’s latent preference and robot’s representation space. RAPL uses human preference rankings over video triplets to learn representations aligned with preferences of humans and it then uses optimal transport to map the aligned representation to the reward for training. Authors present results in simulation on table top manipulation and 2 block rearrangement tasks. They demonstrate RAPL outperforms and is sample efficient compared to RLHF and temporal consistency based methods in all environments. In addition, they also show RAPL enables zero-shot generalization of rewards to new embodiments for both tasks.

**Strengths:**

1. Idea of aligning visual representations to human preferences and using that to extract preference based rewards is interesting and novel
2. Experiments are well designed and demonstrate RAPL outperforms vanilla RLHF in the studied simpler manipulation and block movement tasks is promising.
3. Results demonstrating zero-shot generalization to new embodiements and sample efficiency gains further strengthen benefits of using RAPL.
4. Paper is well written and easy to follow

**Weaknesses:**

1. Experimental setup is promising but the evaluation is done on simpler tasks. With just 150 triplet dataset used for both the tasks to learn preferences the tasks being studied brings up a question about how well this approach works on more complex tasks? For example, the block movement task in x-magical environment only requires agent to avoid the blue box in the environment. This can essentially be treated as a simple obstacle to avoid in the environment which is simple to learn in general. Some example tasks authors could use for evaluation are: block stacking task with constraints where a user prefers stacking blocks based on size i.e. largest block is at lowest followed by smaller blocks irrespective of the color of boxes. This setup tests generalization ability of the policy to different colors and ability to reason about object size based on preferences. My point being preference-based learning needs to be evaluated with tasks that have distractors that are difficult to pick up. It’d be nice if authors can add more experiments on different manipulation tasks with such properties
2. One baseline I’d like to see in addition to RAPL is how well does a simple contrastive pretrained visual representation as reward with same optimal transport logic performs in comparison to RAPL. It is unclear how beneficial RAPL’s triplet sampling is for the current set of tasks being considered.
3. For the cross-embodiment experiments the simple tasks of block movement doesn’t necessarily highlight how well the proposed approach is performing. I’d be interested in seeing similar results in simple object pick and place task similar to one asked in W1.
4. Human preferences are multi-modal in nature i.e. different people have different preferences. The current setup doesn’t consider multi-modal preferences and show experiments comparing different methods in such a setup which seems like a big flaw in current experimental setup. I’d like authors to consider adding experiments that’d demonstrate effectiveness of RAPL under such setting

**Questions:**

1. In Figure 6. the RLHF results are quite poor. Have authors tried scaling training for RLHF for the same experiment to figure out how long does it take to reach RAPL/GT performance?

My major concern is evaluation on simple object movement tasks and no evaluation on tasks with multi-modal preferences if authors add more experiments that'd address my concerns I’d be happy to update the rating

---

> ### Author Response · Authors · 2023-11-22
> **Responses to reviewer sCYJ**
>
> We would like to thank the reviewer for taking the time to review our paper and provide valuable feedback. We are glad that the reviewer thinks our paper is sound and well presented. We are excited to have the chance to address the reviewer’s questions and concerns. These edits will make the paper stronger.
>
> ***Q1&3.  Experiment set up is easy. Experiment setup that tests generalization ability of the policy to different colors and ability to reason about object size based on preferences.***
>
> A1&3. We agree that this is a very interesting suggestion! Per your suggestion, we implemented a more complex robot manipulation task (described in detail in the common response and in the Appendix of the updated manuscript). Specifically, we modified the robot manipulation environment described in subsection 5.2 by 1) increasing the manipulation difficulty of the task (more objects of various colors and sizes, robot needs to grasp the object while pushing), 2) introducing a more complex preference (grouping by object size), and 3) adding visual distractors (object color). Please find the detailed description in Appendix Section A.8. We find that in this more complex manipulation and preference-learning task, our method significantly outperforms all the baselines (including the new baselines as introduced in our common response).
>
> ***Q2. Additional OT-based visual reward that uses contrastive pretrained visual representation.***
>
> A2. We would like to thank the reviewer for the baseline suggestion. We additionally compare our method with OT-based visual reward using R3M representation (Nair et al., 2022), which is a off-the-shelf ResNet-18 encoder pre-trained on the Ego4D data set via a learning objective that combines time contrastive learning and video-language alignment. The results are reported in Appendix Section A.7. This new baseline achieves near 0 success rate. Similar poor results are observed also in the (Haldar et al., 2023b) where an OT-based visual reward with a pre-trained R3M representation model gives near 0 success rate for manipulation tasks.
>
> ***Q4. Human preferences are multi-modal in nature i.e. different people have different preferences. The current setup doesn’t consider multi-modal preferences and show experiments comparing different methods in such a setup which seems like a big flaw in current experimental setup. I’d like authors to consider adding experiments that’d demonstrate effectiveness of RAPL under such setting.***
>
> A4. Indeed, when learning from multiple people, each user could have a different preference than other users. However, in our paper, we focus on a personalized setting in which the robot aims to align its representation and visual reward with a single end-user, i.e., the user’s preference is uni-modal. We are very excited about the proposed extension where preference feedback is crowdsourced from different people (i.e., multi-modal preference), and we included this in the conclusion and future work.
>
> ***Q5. In Figure 6. the RLHF results are quite poor. Have authors tried scaling training for RLHF for the same experiment to figure out how long does it take to reach RAPL/GT performance?***
>
> A5.  We investigated both RLHF performance as a function of training time and as a function of preference dataset size. First, we compared the true success rate of the RLHF policy (under the true human’s measure of success) vs. the “perceived” policy performance under the RLHF reward. These results are visualized in Figure 18 in Appendix Section A.10. We see that after 350 epochs, the RLHF learned reward perceives the policy as achieving a high reward. However, the true success rate is still near zero. This indicates that the RL optimization is capable of improving reward over time, but it is optimizing a poor reward signal that does not correlate with the true measure of success. Second, we investigated if RLHF performance can be improved by scaling the preference dataset. In the last paragraph of Section 6 and Appendix Section A.5, we doubled the training dataset set for training the RLHF reward and tested the policy training with this reward in the Franka manipulation task (Figure 12 in Appendix Section A.5). We observe that policy performance is slightly improved (indicating that with more feedback data, RLHF could yield to an aligned policy), but our method still outperforms RLHF by 75% with 50% less training data. To build intuition for this, we visualized the attention map in Appendix Section A.4. We can observe that the RLHF embedding pays attention to irrelevant areas that can induce spurious correlations, which may also contribute to the poorer performance of the RLHF visual reward.

---

> > ### Author Response · Authors · 2023-11-22
> >
> > Dear reviewer sCYJ,
> >
> > We would like to express our appreciation for your valuable feedback and suggestions!
> >
> > We are wondering if our additional experiment results on the suggested baselines in the new complex environment, quantitative analysis on the poor performance of RLHF, and explanations for using uni-modal human preference address your comments about the paper? If you have further questions, we would be happy to provide further information!
> >
> > Best regards,
> >
> > Authors

---

### Official Review · Reviewer_NUZE · 2023-11-01

**Soundness:** 3 good
**Presentation:** 4 excellent
**Contribution:** 3 good
**Rating:** 6
**Confidence:** 3

**Summary:**

The authors proposes Representation Aligned Preference-based Learning (RAPL).
It is a video-only tractable method to use human feedback preferences between three trajectories to align visual representations with what matter to the tasks.

RAPL first asks humans to rate preferences over a set of video triplets.

It then uses the Bradley-Terry model to interpret and leverage the preferences, but not directly to reward prediction.
Instead, it focuses all the data on representation alignment on the video pre-trained visual features (ResNet18).
It uses the optimal transport method, which transports one video embedding distribution to another, with minimal cost.
Given the aligned representation RAPL then uses optimal transport to design a visual reward for the robot policy.

RAPL is then tested against ground truth policy, TCC, and vanilla RLHF.
We first tested the same embodiment for both training and testing.
This first test is done on a toy X-MAGICAL environment, and then a realistic IsaacGym simulator.
For the former, the task is kitchen top cleaning, with avoiding off-limits zone.
With the triplet training size of 150, RAPL is able not only to make spatial progress, but to be close to human preference.
This results in a higher overall binary success rate on the task.

The IsaacGym simulator also shows similar results, where the RAPL features are aligned with GT on preferred videos, and no alignment on dislikes.
Further analysis shows that RAPL focuses on task-relevant objects and regions is the reason for the significantly higher final task success rate.

RAPL is then tested on zero-shot generalization setup, where the testing robot embodiment is different from that of training.
This second test is also done in X_MAGICAL, and then the robot simulator. For both RAPL is the best, closest to the ground truth performance.
In fact, on some combination (X-MAGICAL train on short stick, test on medium stick), RAPL improves its performance of the previous non-cross modal setup.

**Strengths:**

The paper presented a novel preference-based representation alignment and robot policy learning techniques.

The testing section is quite thorough and goes in-depth on the various nuances of what is being learned and accomplished in the actual robot task.

**Weaknesses:**

No discussion on what is still hard to do or not reliable.
Also, evaluation based on difficulty level of the configuration, shape, number of objects needed to be cleared would help toward answering the previous question.
Another is to compare with a second, very different, task.

**Questions:**

How does the visual attention map (figure 11, supplementary material) computed?

---

> ### Author Response · Authors · 2023-11-22
> **Responses to reviewer NUZE**
>
> We would like to thank the reviewer for taking the time to review our paper and provide valuable feedback. We are glad that the reviewer thinks our paper is sound and well presented. We are excited to have the chance to address the reviewer’s questions and concerns. These edits will make the paper stronger.
>
> ***Q1.  No discussion on what is still hard to do or not reliable.***
>
> A1. We have expanded our conclusion section to reflect the limitations of our method. One limitation of the method is that we assume the user is rational, which motivates the usage of the Bradley-Terry model for interpreting the preference feedback. Although such an assumption is widely used in the preference learning community, humans can exhibit irrationality, systematic bias, and suboptimality. Future study can focus on the open problem of robustly aligning visual representations under such forms of feedback. Moreover, although our method is much more sample efficient compared to the vanilla RLHF, it still passively learns from the human feedback which makes it still inefficient for product deployment. Future work could investigate the challenging problem of actively querying humans for faster aligning visual representations.
>
> ***Q2. Also, evaluation based on difficulty level of the configuration, shape, number of objects needed to be cleared would help toward answering the previous question. Another is to compare with a second, very different, task.***
>
> A2. We would like to thank the reviewer for your suggestion! Per your suggestion, we implemented a more complex robot manipulation task (described in detail in the common response and in the Appendix of the updated manuscript). Specifically, we modified the robot manipulation environment described in subsection 5.2 by 1) increasing the manipulation difficulty of the task (more objects of various colors and sizes, robot needs to grasp the object while pushing), 2) introducing a more complex preference (grouping by object size), and 3) adding visual distractors (object color). Please find the detailed description in Appendix Section A.8. We find that in this more complex manipulation and preference-learning task, our method significantly outperforms all the baselines  (including the new baselines as introduced in our common response).
>
> ***Q3. How is the visual attention map computed?***
>
> A3. We added a detailed explanation and graphical visualization of how we compute the attention map in Appendix Section A.4. In short, we visualize the attention map by leveraging linear permutation and kernel inflation.

---

> > ### Comment · Reviewer_NUZE · 2023-11-22
> >
> > Dear authors,
> > Thank you so much for the additional information. After reading them, as well as other discussions, I have decided to stay with the current score, albeit a more solid score.

---

> > > ### Author Response · Authors · 2023-11-22
> > >
> > > Dear reviewer NUZE,
> > >
> > > We are glad that our responses could address your questions, and we appreciate your positive feedback!
> > >
> > > Thank you again for taking the time to review our paper and providing insightful feedback!
> > >
> > > Best regards,
> > >
> > > Authors

---

### Author Response · Authors · 2023-11-22
**General statement**

We thank all the reviewers for their helpful comments and suggestions. We are happy that the reviewers thought our idea of aligning visual representations using human preferences and using that to extract preference based rewards is interesting and novel, and our paper is well-presented. We appreciate this opportunity to address some questions and make improvements to the manuscript, which are highlighted in the manuscript in blue. Specifically, we made the following major changes:

**1. More baselines to compare with our proposed visual reward.** Per the reviewer’s suggestions, we added three additional OT-based reward baselines to compare with our proposed approach in the robot manipulation setting: (1) *Fine-Tuned-MVP-OT*, which fine-tunes MVP representation using images from the task environment; (2) *R3M-OT*, which is an off-the-shelf ResNet-18 encoder pre-trained on the Ego4D data set via a learning objective that combines time contrastive learning, video-language alignment, and a sparsity penalty (Nair et al., 2022); (3) *ImageNet-OT*, which is a ResNet-18 encoder pre-trained on ImageNet. We find that our method significantly outperforms these new baselines in terms of task success, and detailed results can be found in Appendix Section A.7.

**2. Additional experiments on a more complex robot manipulation task.** Per the reviewer’s suggestion, we implemented a more complex robot manipulation task to further validate RAPL’s ability to disentangle visual features that underlie an end-users preferences. We increase the difficulty of the robot manipulation environment described in Section 5.2 by introducing more objects to the tabletop as well as adding visual distractors that are irrelevant to the human's preferences. The environment has multiple objects on the table of various colors---red, green, and “goal-region”-blue---and some of the objects are cubes while others are rectangular prisms. The Franka robot arm needs to learn that the end-user prefers to push the *rectangular* objects (instead of the cube objects) *efficiently together* (instead of one-at-a-time) to the goal region. Compared to the original manipulation task described in Section 5.2, *the Franka arm needs to learn representations that can disentangle both semantic preference (grouping) and low-level preference (shape) from RGB images under visual distractors (object color)*. In addition, the Franka arm needs to learn to grasp the rectangular object first and use that to push the second rectangular object, as it is difficult to directly push the rectangular object stably using the robot finger gripper, thus increasing the task difficulty. The detailed task description and results are reported in Appendix Section A.8. We find that in this more complex manipulation and preference-learning task, our method significantly outperformed all the baselines.

**3. Quantitative analysis of the relationship between the learned visual reward and the end-user’s ground-truth reward.** We thank the reviewer for their helpful suggestion of computing the relationship between the learned and ground-truth reward. For each robot manipulation task, we computed the average Spearman’s correlation coefficient across 100 video trajectories. We found that our learned visual reward shows a stronger correlation to the GT reward compared to baselines. We added the detailed results in Appendix Section A.9.

|              | Franka Group (Sec. 5.2) | Kuka Group (Sec. A.6) | Franka Clutter (Sec. A.8) |
|--------------|-------------------------|-----------------------|---------------------------|
| **RAPL**     | 0.59                    | 0.47                  | 0.61                      |
| **RLHF**     | 0.38                    | 0.31                  | 0.26                      |
| **MVP-OT**   | -0.1                    | 0.02                  | 0.08                      |
| **Fine-Tuned-MVP-OT** | 0.19            | 0.02                  | 0.11                      |
| **ImageNet-OT** | -0.09                | 0.12                  | -0.02                     |
| **R3M-OT**   | 0.03                    | -0.14                 | -0.17                     |

Table 1: Spearman’s rank correlation coefficient between the GT reward and each learned reward.

We look forward to continued discussion with the reviewers about our updated manuscript.

---

### Meta-Review · Area_Chair_eB2F · 2023-12-10

**Metareview:**

The paper proposes Representation-Aligned Preference-based Learning (RAPL) method to align robot’s visual representation using human preferences.  This is achieved by learning representations aligned with human preference rankings over video triplets and mapping the representation to reward via optimal transport. Experiments on simulation on table top manipulation and 2 block rearrangement tasks are used to evaluate the method.

Most of the reviewers rate the papers favorable (6,6,6) except one (3).  The biggest criticism of the critical reviewer includes missing or weak baselines and that the experiments are in simulation. Authors provide a detailed rebuttal, to which there is no further response from the reviewer. The rebuttal provides an in-depth answers to the questions raised by the reviewer and given the supportive reviews from other 3 reviewers I recommend accepting the paper.

**Justification For Why Not Higher Score:**

Most of the reviewers are mildy positive.

**Justification For Why Not Lower Score:**

The rebuttal addresses most of the criticism raised by the most critical reviewer.

---

### Decision · Program_Chairs · 2024-01-16

Accept (poster)